# Soil-transmitted helminth infections and nutritional status in Ecuador: findings from a national survey and implications for control strategies

Ana L Moncayo,[1] Raquel Lovato,[2] Philip J Cooper[3,4]

[1]Centro de Investigación Para la Salud en América Latina (CISeAL), Pontificia Universidad Católica del Ecuador, Quito, Pichincha, Ecuador
[2]Dirección Nacional de Vigilancia Epidemiológica, Ministerio de Salud Pública del Ecuador, Quito, Pichincha, Ecuador
[3]Facultad de Ciencias Médicas, de la Salud y la Vida, Universidad Internacional del Ecuador, Quito, Ecuador
[4]Institute of Infection and Immunity, St George's University of London, London, UK

**Correspondence to**
Dr Ana L Moncayo;
amoncayo708@puce.edu.ec

## ABSTRACT

**Objective** The estimation of prevalence and intensity of soil-transmitted helminth (STH) infections at a country-level is an essential prerequisite for the implementation of a rational control programme. The aim of this present study was to estimate the prevalence and distribution of STH infections and malnutrition in school-age children in rural areas of Ecuador.

**Design** Cross-sectional study from October 2011 to May 2012.

**Setting** Eighteen rural schools were randomly selected from the three ecological regions of Ecuador (coastal, highlands and Amazon basin).

**Participants** 920 children aged 6–16 years.

**Main outcome measures** Prevalence and intensity of STH infections associated with malnutrition (thinness/wasting or stunting).

**Results** The results showed that 257 (27.9%) children were infected with at least one STH parasite. The prevalence of *Trichuris trichiura*, *Ascaris lumbricoides* and hookworm was 19.3%, 18.5% and 5.0%, respectively. Malnutrition was present in 14.2% of children and most common was stunting (12.3%). Compared with other regions, schoolchildren in the Amazon region had the highest STH prevalence (58.9%) of which a greater proportion of infections were moderate/heavy intensity (45.6%) and had the highest prevalence of malnutrition (20.4%). A positive association was observed between moderate to heavy infections with *A. lumbricoides* and malnutrition (adjusted OR 1.85, 95% CI 1.04 to 3.31, p=0.037).

**Conclusions** Our estimate of the prevalence of STH infections of 27.9% at a national level in Ecuador is lower than suggested by previous studies. Our data indicate that schoolchildren living in the Amazon region have a greater risk of STH infection and stunting compared with children from other regions. The implementation of school-based preventive chemotherapy and nutritional supplement programmes within the Amazon region should be prioritised. Long-term control strategies require improvements in water, sanitation and hygiene.

## INTRODUCTION

Soil-transmitted helminth (STH) infections (*Ascaris lumbricoides*, *Trichuris trichiura* and hookworm) are a major public health

### Strengths and limitations of this study

► This study is the first national survey of soil-transmitted helminth (STH) infections in schoolchildren in Ecuador and will inform the strategy for a national plan for deworming of schoolchildren in the three ecological regions in Ecuador.

► These baseline data will allow health authorities to monitor the impact of future deworming programmes among schoolchildren.

► The sampling strategy may not have detected localised geographical areas within the three regions with a higher prevalence of STH infections.

► The cross-sectional nature of the study does not allow us to determine the temporal relationship between the STH infections and malnutrition.

► Lack of a more complete nutritional evaluation including micronutrients.

problem in tropical and subtropical regions of low-income and middle-income countries, especially in marginalised population with poor access to clean water and sanitation, and living in overcrowded conditions with low levels of education and lack of access to health services.[1] Estimates suggest that more than 1.45 billion humans worldwide are infected with STH parasites causing up to 4.98 million years lost due to disability and 5.18 disability-adjusted life years.[2] Children are the group at highest risk of infection and most vulnerable to the pathological consequences of infection. An estimated 13.9 million preschool and 35.4 million schoolchildren in Latin America and the Caribbean (LAC) were at risk of infection by STH in 2012.[3] STH infections are considered to have important deleterious effects on the nutritional status, growth and physical development of infected children,[4 5] and may also affect cognitive performance and educability.[5]

In the last decade, an increasing number of international initiatives have established the aim to either reduce or to eliminate the disease

burden caused by STHs and other helminth parasites prevalent in resource-poor regions. Chemotherapy still remains the most effective short-term approach for STH control in areas where infections are highly endemic. The World Health Assembly in 2001 urged all member states where STH infections are endemic to treat at least 75% and up to 100% of all school-age children at risk of morbidity by 2010.[6] In Ecuador, international agencies and non-governmental organisations have formed partnerships with the government to provide deworming treatments for a large number of schoolchildren and achieved the goal of at least 75% coverage of eligible schoolchildren in 2006 and 2009, years for which data are available.[7] However, since 2009, there are no available data on treatment coverage of preschool and school-age children.

Few countries in LAC have implemented nationwide surveys on prevalence and intensity of STH infections in order to plan treatment strategies. A recent study identified gaps in available data on STH infections using data published between 2000 and 2010 in LAC.[8] A total of 335 published studies of STH prevalence were found in 18 countries: in Ecuador, 11 studies were analysed, of which two estimated a prevalence below 20%, four (36.4%) estimated a prevalence between 20% and 50% and five (45.5%) a prevalence of >50%. These data come from highly focal studies restricted to one or more parishes in 7 of the country's 24 provinces.

The aim of the present study was to carry out a national survey to estimate the current prevalence and intensity of STH infections and malnutrition in school-age children. These data will be used to design and evaluate appropriate intervention strategies within the country.

## METHODS

### Study area and population

The study was conducted in schoolchildren living in three distinct ecological regions of Ecuador: the Andes highlands, the coastal lowlands and the Amazon plains. The coastal region consists of six provinces covering approximately $70\,000\,km^2$ constituting less than a third of the surface area of Ecuador, but where 50% of the population lives.[9] The average annual temperature is between 24°C and 26°C (18°C–30°C) and the hottest period occurs during the rainy season from February to April. The climate is greatly influenced by the ocean currents, 'El Niño' (warm) and 'Humboldt' (cold). The land is generally low-lying with elevations below 800 m above sea level (masl).[10 11] The Andean region of 11 provinces bisects the country from North to South, covering about one-fifth of land surface of Ecuador and is inhabited by 44.5% of the population.[9] In this region, the altitude ranges from 1200 to 6000 masl, and daily temperature varies between 3°C and 25°C depending on altitude. The dry season is between July and August, and periods of high rainfall are generally seen between March and April and again in October.[10 11] The Amazon region consists of six provinces and covers an area of $120\,000\,km^2$ or about 50%

of the land surface but is where only 5.1% of the population lives.[9] This region is characterised by an average of about 3000 mm of rainfall per year and an average annual temperature of 23°C, high humidity and relatively constant rainfall throughout the year.[10 11]

Poverty based on unsatisfied basic needs was higher in the Amazon region (51.7%) than in the other regions (coastal region: 40.1% and Andean region: 20.7%) in 2016.[12] Net enrolment on basic education and illiteracy rates were higher in Andean region (97.4%, 5.9%) than in coastal (95.4%, 5.5%) and Amazon regions (95.7%, 5.0%).[12]

### Study design and sample size

A cross-sectional study was conducted to estimate the prevalence and intensity of STH infections among rural schoolchildren at a national level. The study design followed WHO recommendations to allow comparison with other international studies.[13] The three ecological regions constituted the strata for sampling. WHO recommendation is that 200–250 individuals in each zone or strata should be adequate to assess the need for control measures.[13] A list of all primary rural schools was compiled by the National Education Officer, and six schools in each strata were selected at random. Private rural schools were included in the sampling although the vast majority was free-access government-funded schools. Fifty children attending the fifth grade (11–12 years of age) of primary education and who were present on the day of the survey were randomly selected in each school using a list of children provided by the school principal. If 50 were not present, we randomly selected children from other grades to complete the required number. Thus, overall we aimed to sample 300 children in each ecological zone to provide a total sample of 900 children.

### Data collection

The field study was conducted between October 2011 and May 2012. Two data-collection forms were adapted from WHO guidelines[13] to collect demographic, parasitological and anthropometric data from children and information about access to clean water and sanitation in schools, distance to healthcare services and recent anthelmintic treatments provided in the schools. Information on water supply, sanitation and anthelmintic treatment in schools was collected from teachers through an investigator-administered questionnaire and confirmed by direct observation.

### Examination of stool samples

Single stool samples were collected from each child in plastic containers and examined for eggs using the standard Kato-Katz technique for STHs.[14] Kato-Katz slides were examined within 30–60 min of preparation. STH prevalence was expressed as the percentage of children found positive for each parasite and the prevalence of infection with at least one STH parasite. The number of eggs per gram of faeces (epg) was calculated by multiplying the egg count obtained by Kato-Katz by a conversion factor of 24.

WHO guidelines were used to classify children as having light, moderate and heavy infections with each parasite as follows[15]: for *A. lumbricoides*, light (1–4999 epg), moderate (5000–49 999 epg) and heavy (≥50 000 epg); for *T. trichiura*, light (1–999 epg), moderate (1000–9999 epg) and heavy (≥10 000 epg); and for hookworm, light (1–1999 epg), moderate (2000–3999 epg) and heavy (≥4000 epg).

### Anthropometric measurements

Height was measured without shoes using a portable stadiometer and weight with a digital scale (Filizola, E-150/3P model). The instruments were calibrated periodically. All measurements were performed by trained personnel. Height-for-age z scores (HAZ) and Body mass index-for-age z scores (BAZ) were calculated using WHO Child Growth Standards.[16] Children with HAZ and BAZ <-2 SD were classified as stunted and thin/wasted, respectively. Children were classified as having malnutrition if they had at least one of these two conditions.

### Statistical analysis

Data were double entered into Epi Info V.7 and then exported into STATA V.10 for analysis. The prevalence, intensity of infection and 95% confidence intervals (CIs) were estimated for each ecological zone. The weighted prevalence for any helminth was also calculated using the number of children aged 5–14 years in rural areas of each region. In a bivariate analysis, the $\chi^2$ test or Fisher's exact test (p<0.05) was used to compare STH prevalence and infection intensity groups between ecological zones. In multivariable analysis, the association between geohelminth infections and malnutrition was assessed by logistic regression analysis with random effects to obtain robust SEs taking into account the effect of clustering by school. Odds ratios (OR) and 95% CIs were estimated. Confounding factors controlled in the analysis included sex, age, waste water disposal system, drinking water and recent anthelmintic treatment.

### Patient and public involvement

Patients were not involved in the design, and organisation of recruitment and conduct of the study. The parent of each child received the results of the parasitological examination.

## RESULTS

### Characteristics of study population

A total of 938 children from the three regions and 18 schools studied were eligible, of which 920 (98.0%) were recruited and provided stool samples. Table 1 shows the characteristics of the study population. The mean age was 10.3 years (range 6–16), and there were slightly more girls than boys (51.2% vs 48.8%). Children who participated in the study attended mostly Spanish-speaking schools (79.5%). Access to piped water (52.4%) and a public sewer system (50.4%) in schools was present for approximately half the children evaluated. Most of the children

**Table 1** Characteristics of school-age children sampled in Ecuador, 2011–2012 (n=920).

| Variables | N | % |
|---|---|---|
| **Age (years)** | | |
| 6–10 | 450 | 48.9 |
| 11–16 | 470 | 51.1 |
| **Sex** | | |
| Male | 449 | 48.8 |
| Female | 471 | 51.2 |
| **Rural schools** | | |
| Public-Spanish speaking | 731 | 79.5 |
| Public/municipal-Spanish | 30 | 3.3 |
| Municipal-Spanish | 66 | 7.2 |
| Public-bilingual* | 49 | 5.3 |
| Private-Spanish | 44 | 4.8 |
| **Drinking water at schools** | | |
| Piped | 482 | 52.4 |
| Well | 105 | 11.4 |
| River or stream | 333 | 36.2 |
| **Sewage at schools** | | |
| Public sewer system | 464 | 50.4 |
| Septic tank | 379 | 41.2 |
| No sewage system | 77 | 8.4 |
| **Distance from school to a health service** | | |
| ≤1 km | 649 | 70.5 |
| >1 km | 271 | 29.5 |
| **Last anthelmintic treatment in school** | | |
| ≤6 months | 208 | 22.6 |
| 7–12 months | 568 | 61.7 |
| >12 months/none | 144 | 15.7 |
| **Anthelmintic drug given in school** | | |
| Albendazole | 625 | 67.9 |
| Unknown | 295 | 32.1 |
| **School location (altitude)** | | |
| 0–207 masl | 345 | 37.5 |
| 243–1869 masl | 315 | 34.2 |
| ≥2345 masl | 260 | 28.3 |

*Bilingual—Spanish and Quechua.
masl, metres above sea level.

had received an anthelmintic drug during the previous 12 months (84.3%), and 70.5% had a health facility near their school (ie, <1 km away).

### Prevalence and intensity of STH infection

The overall prevalence of infection with at least one of the STH parasites was 27.9% (95% CI 25.0 to 30.8), and the weighted prevalence was 17.6%. Most frequent infections were with *T. trichiura* (19.3%, 95% CI 16.8 to 21.9) followed by *A. lumbricoides* (18.5%, 95% CI 16.0 to 21.0) and hookworm (5.0%, 95% CI 3.6 to 6.4).

**Table 2** Prevalence and intensity of soil-transmitted helminths in schoolchildren in Ecuador by region, 2011–2012.

| | Coast, n=303 | | Highlands, n=308 | | Amazon, n=309 | | Total, n=920 | |
|---|---|---|---|---|---|---|---|---|
| | n | % (95% CI) | n | % (95% CI) | n | % (95% CI) | n | % (95% CI) |
| Prevalence of infection | | | | | | | | |
| Any geohelminth*** | 44 | 14.5 (0.1 to 18.5) | 31 | 10.1 (6.7 to 13.4) | 182 | 58.9 (53.0 to 64.0) | 257 | 27.9 (25.0 to 30.8) |
| *Ascaris lumbricoides*** | 13 | 4.3 (2.0 to 6.6) | 23 | 7.5 (4.5 to 10.4) | 134 | 43.4 (37.8 to 48.9) | 170 | 18.5 (16.0 to 21.0) |
| *Trichuris trichiura*** | 36 | 11.9 (8.2 to 15.5) | 10 | 3.2 (1.3 to 5.2) | 132 | 42.7 (37.2 to 48.3) | 178 | 19.3 (16.8 to 21.9) |
| Hookworms*** | 0 | 0 | 1 | 0.3 (0 to 0.9) | 45 | 14.6 (10.6 to 18.5) | 46 | 5.0 (3.6 to 6.4) |
| Mixed infections*** | | | | | | | | |
| One parasite | 39 | 88.6 (78.9 to 98.4) | 29 | 93.6 (84.4 to 100) | 81 | 44.5 (37.2 to 51.8) | 149 | 58.0 (51.9 to 64.1) |
| Two parasites | 5 | 11.4 (1.6 to 21.1) | 1 | 3.2 (0 to 9.8) | 73 | 40.1 (32.9 to 47.3) | 79 | 30.7 (25.1 to 36.4) |
| Three parasites | 0 | 0 | 1 | 3.2 (0 to 9.8) | 28 | 15.4 (10.1 to 20.7) | 29 | 11.3 (7.4 to 15.2) |
| Intensity of infection | | | | | | | | |
| Any helminth*** | | | | | | | | |
| Light | 37 | 84.1 (73.1 to 95.1) | 24 | 77.4 (62.4 to 92.5) | 99 | 54.4 (47.1 to 61.7) | 160 | 62.3 (56.3 to 68.2) |
| Moderate | 6 | 13.6 (3.3 to 23.9) | 7 | 22.6 (7.5 to 37.6) | 57 | 31.3 (24.5 to 38.1) | 70 | 27.2 (21.8 to 32.7) |
| Heavy | 1 | 2.3 (0 to 6.7) | 0 | 0 | 26 | 14.3 (9.2 to 19.4) | 27 | 10.5 (6.7 to 14.3) |
| Moderate to heavy | 7 | 15.9 (6.6 to 30.1) | 7 | 22.6 (7.5 to 37.6) | 83 | 45.6 (38.2 to 53.1) | 97 | 37.7 (31.8 to 44.0) |
| *A. lumbricoides*** | | | | | | | | |
| Light | 9 | 69.2 (42.9 to 95.5) | 16 | 69.6 (50.2 to 88.9) | 54 | 40.3 (31.9 to 48.7) | 79 | 46.5 (38.9 to 54.0) |
| Moderate | 4 | 30.8 (4.5 to 57.1) | 7 | 30.4 (11.1 to 49.8) | 55 | 41.0 (32.6 to 49.5) | 66 | 38.8 (31.4 to 46.2) |
| Heavy | 0 | 0 | 0 | 0 | 25 | 18.7 (12.0 to 25.3) | 25 | 14.7 (9.3 to 20.1) |
| *T. trichiura** | | | | | | | | |
| Light | 33 | 91.7 (82.4 to 100.9) | 10 | 100 | 105 | 79.5 (72.6 to 86.5) | 148 | 83.1 (77.6 to 88.7) |
| Moderate | 2 | 5.6 (0 to 13.2) | 0 | 0 | 27 | 20.5 (13.5 to 27.4) | 29 | 16.3 (10.8 to 21.8) |
| Heavy | 1 | 2.7 (0 to 8.3) | 0 | 0 | 0 | 0 | 1 | 0.6 (0 to 1.7) |
| Hookworms | | | | | | | | |
| Light | 0 | 0 | 1 | 100 | 41 | 91.1 (82.5 to 99.7) | 42 | 91.3 (82.8 to 99.8) |
| Moderate | 0 | 0 | 0 | 0 | 3 | 6.7 (0 to 14.2) | 3 | 6.5 (0 to 13.9) |
| Heavy | 0 | 0 | 0 | 0 | 1 | 2.2 (0 to 6.7) | 1 | 2.2 (0 to 6.5) |

* p<0.05, **p<0.01, ***p <0.001.

Forty-two per cent of infected children had polyparasitism (ie, two or three parasites) (table 2). The overall prevalence of moderate-intensity and high-intensity STH infections among infected children was 37.7% (95% CI 31.8 to 44.0), with the greatest proportions observed in the Amazon region (45.6%, 95% CI 38.2 to 53.1) and primarily consisting of infections with *A. lumbricoides* (table 2).

Age prevalence and intensity patterns for *A. lumbricoides* and *T. trichiura* did not vary markedly by age in this sample of children aged 6–16 years although prevalence of both parasites was greater in older children (online supplementary figure S1). Neither prevalence nor intensity of infection differed significantly by sex for the two parasites (data not shown).

### Prevalence and intensity of STH infections by region

Of the 18 selected schools, six were located in the Amazon region (n=309 children), six in the highland region (n=308 children) and six in the coastal region (n=303 children). Significant differences in prevalence (p<0.001, for all STH infections) and intensity (p<0.01 for *A. lumbricoides* and p<0.05 for *T. trichiura*) were observed between the three regions. The highest rates of *A. lumbricoides, T. trichiura* and hookworm infection were in the Amazon region (43.4%, 42.7% and 14.6%, respectively). The other two regions showed prevalence rates varying between 0% and 11.9% for each of the three parasites. Among infected children, heavy infection intensities with *A. lumbricoides* (18.7%, 95% CI 12.0 to 25.3) and hookworm (2.2%, 95% CI 0 to 6.7) were

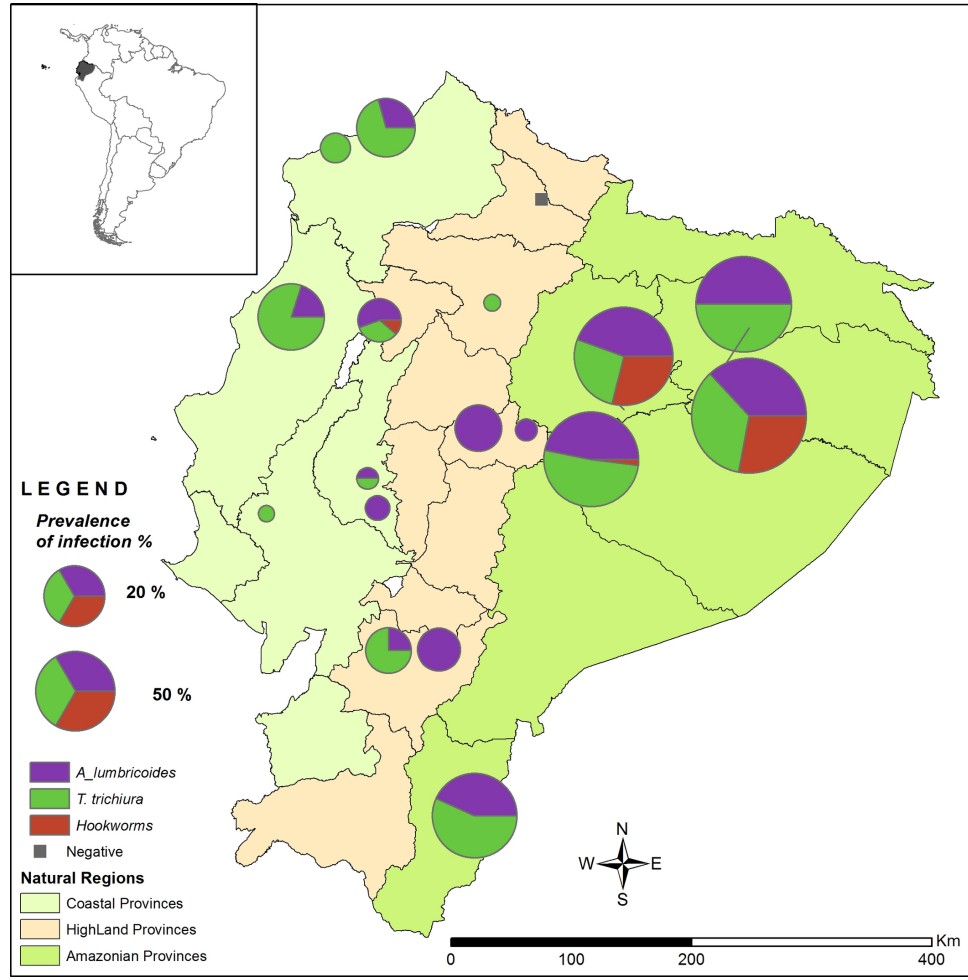

**Figure 1** Prevalence of soil-transmitted helminth infections in three ecological regions of Ecuador. Location of each school surveyed is marked. The size of the symbol corresponds to the % of children infected with STH in each school. Colours represent the three species found: Purple=*Ascaris lumbricoides*, green=*Trichuris trichiura* and red=hookworms.

only detected in the Amazon region. In contrast, children with heavy infection intensities with *T. trichiura* were only found in the coastal region (2.7%, 95% CI 0 to 8.3) (table 2 and figure 1).

### Prevalence of malnutrition

Table 3 summarises the anthropometric findings of the children surveyed. The overall prevalence of stunting (HAZ <-2 SD) and thinness/wasting (BAZ <-2 SD) was 12.3% (95% CI 10.2% to 14.4%) and 2.1% (95% CI 1.1% to 3.0%), respectively. The percentage of children with malnutrition (at least one of the two conditions) was 14.2% (95% CI 11.98% to 16.50%). The prevalence of stunting varied significantly by region (p<0.001) with the highest prevalence observed in the Amazon region

**Table 3** Nutritional and anthropometric characteristics of sampled Ecuadorian schoolchildren by region, 2011–2012

|  | Total | Coast | Highlands | Amazon |  |
|---|---|---|---|---|---|
|  | n=920 | n=303 | n=308 | n=309 | P values* |
| Mean BAZ (range) | 0.31 (-3.43–4.34) | 0.32 (-3.43–4.34) | 0.29 (-3.41–3.19) | 0.31 (-2.73–3.47) | 0.960 |
| Mean HAZ (range) | −0.79 (-6.95–2.89) | −0.41 (-4.25–2.88) | −0.73 (-3.58–2.89) | −1.22 (-6.95–1.78) | <0.001 |
| Wasting (%)† | 19 (2.1) | 10 (3.3) | 4 (1.3) | 5 (1.6) | 0.175 |
| Stunting (%)‡ | 113 (12.3) | 18 (5.9) | 37 (12.0) | 58 (18.8) | <0.001 |
| Malnutrition (%)§ | 131 (14.2) | 27 (8.9) | 41 (13.3) | 63 (20.4) | <0.001 |

*P value refers to significance of mean and prevalence differences between the three regions by analysis of variance or $X^2$ testing.
†Wasting: BAZ <-2 SD.
‡Stunting: HAZ<-2 SD.
§Malnutrition: at least one of thinness/wasting or stunting.
BAZ, body mass index-for-age z score; HAZ, height-for-age z score.

**Table 4**  Bivariate and multivariable analyses of the associations between STH infection and malnutrition* among schoolchildren in Ecuador, 2011–2012

| STH infection | Total n=920 | Malnutrition n (%) | OR (95% CI) Crude | OR (95% CI)† Adjusted | P values |
|---|---|---|---|---|---|
| **Prevalence of infection** | | | | | |
| Any helminth | | | | | |
| Negative | 663 | 78 (11.8) | 1.0 | 1.0 | |
| Positive | 257 | 53 (20.6) | 1.94 (1.33 to 2.86) | 1.38 (0.88 to 2.14) | 0.157 |
| *Ascaris lumbricoides* | | | | | |
| Negative | 750 | 94 (12.5) | 1.0 | 1.0 | |
| Positive | 170 | 37 (21.8) | 1.94 (1.27 to 2.97) | 1.37 (0.86 to 2.20) | 0.190 |
| *Trichuris trichiura* | | | | | |
| Negative | 742 | 93 (12.5) | 1.0 | 1.0 | |
| Positive | 178 | 38 (21.4) | 1.89 (1.25 to 2.88) | 1.29 (0.79 to 2.10) | 0.306 |
| Hookworms | | | | | |
| Negative | 874 | 120 (13.7) | 1.0 | 1.0 | |
| Positive | 46 | 11 (23.9) | 1.97 (0.98 to 3.99) | 1.72 (0.84 to 3.51) | 0.135 |
| Intensity of infection | | | | | |
| *A. lumbricoides* | | | | | |
| None | 750 | 94 (12.5) | 1.0 | 1.0 | |
| Light | 79 | 13 (16.5) | 1.37 (0.73 to 2.59) | 1.04 (0.54 to 2.02) | 0.896 |
| Moderate/heavy | 91 | 24 (26.4) | 2.50 (1.49 to 4.18) | 1.85 (1.04 to 3.31) | 0.037 |
| *T. trichiura* | | | | | |
| None | 742 | 93 (12.5) | 1.0 | 1.0 | |
| Light | 148 | 29 (19.6) | 1.70 (1.07 to 2.69) | 1.19 (0.71 to 2.01) | 0.506 |
| Moderate/heavy | 30 | 9 (30.0) | 2.99 (1.33 to 6.73) | 1.87 (0.76 to 4.60) | 0.173 |
| Hookworms | | | | | |
| None | 874 | 120 (13.7) | 1.0 | 1.0 | |
| Light | 42 | 9 (21.4%) | 1.71 (0.80 to 3.67) | 0.76 (0.32 to 1.79) | 0.526 |
| Moderate/heavy | 4 | 2 (66.7%) | 6.28 (0.88 to 45.03) | 2.58 (0.33 to 20.18) | 0.366 |

*Any of thinness/wasting or stunting.
†OR adjusted by sex, age, sewage, drinking water, anthelmintic treatment and clustering by school.
STH, soil-transmitted helminth.

(18.8%, 95% CI 14.4% to 23.1%). Children living in the coastal region had the highest prevalence of thinness/wasting (3.3%, 95% CI 1.3% to 5.3%); however, the differences between the three regions were not statistically significant (p=0.175) (table 3).

### Association between STH infections and malnutrition

The results of bivariate and multivariable logistic regression analyses of the associations between STH infection and malnutrition are shown in table 4. In bivariate analysis, the prevalence of both *A. lumbricoides* and *T. trichiura* were significantly associated with malnutrition (*A. lumbricoides*: 1.94, 95% CI 1.27 to 2.97; *T. trichiura*: 1.89, 95% CI 1.25 to 2.88), but were no longer significant after adjustment for potential confounders. Hookworm infection did not show a significant association

with malnutrition in either bivariate or multivariable analysis.

There was some evidence of a dose–response relationship such that children with moderate-intensity to heavy-intensity infections with *A. lumbricoides* and *T. trichiura* were more likely to have malnutrition than those children without infections (table 4). However, only the association between moderate to heavy infection intensities with *A. lumbricoides* and malnutrition remained significant in multivariable analyses (adjusted OR 1.85, 95% CI 1.04 to 3.31).

### DISCUSSION

This study provides baseline data on the prevalence and intensity of STH infection and nutritional status in

rural schoolchildren in the three ecological regions of Ecuador. In this population, the overall prevalence of infection for at least one of the STH parasites was less than 50%, and most of the infected children experienced light-intensity infections. *T. trichiura* was the most prevalent parasite (19.3%) followed by *A. lumbricoides* (18.5%) and hookworm (5.0%). The present study found significant regional differences in the prevalence and intensity of STH infections. The greatest prevalence of infection for each of the STH investigated was observed in the Amazon region where about half of the infected children had moderate-to-heavy infection intensities with *A. lumbricoides*. Malnutrition was observed in 14.2% of all children studied with the greatest prevalence seen in the sample from the Amazon region (20.4%).

A systematic review of studies of prevalence of STH infection in South America from 2005 to 2012 estimated a prevalence of 28.1% for Ecuador, similar to the prevalence found in our study. Prevalence rates below 20% were reported for Argentina (18.9%) and Uruguay (18.8%) whereas the highest prevalence rates were reported for French Guyana (46.2%) and Surinam (40.1%). The remaining countries showed prevalence rates of between 25% and 39%.[17]

The high prevalence of STH infection in the Amazon region (58.9%) is comparable with the findings of other studies done in rural areas of the Ecuadorian Amazon that showed a prevalence of between 33.2% and 53.8%.[18–20] A study using a risk index for STH infection (based on census data such as overcrowding and lack of education and sanitation) to estimate STH prevalence within countries, showed high risk and estimated prevalence (ie >50%) in the Amazon basin of Ecuador whereas prevalence outside these high-risk pockets was estimated to be between 20% and 50%.[21] The concordance of observed and estimated prevalence in the Amazon region is unsurprising, considering that the Ecuadorian Amazon is the poorest region in Ecuador. In 2016, poverty levels (based on unsatisfied basic needs) in the Amazon region were 51.7%, well above the national poverty rate of 32.0%.[12] According to the Atlas of Socio-economic Inequalities of Ecuador,[22] the rural areas of the Amazon region are one of the areas in the country in which social conditions are considered to be critical. This area is characterised by deficiencies in housing and health indicators: one out of five households has safe water, and one out of three has adequate walls. Chronic malnutrition affects 27% of children aged 0–60 months.[23] These indicators confirm that STH infections are infections of poverty. Further, environmental and climatic conditions in the Amazon region are likely to be most favourable for the transmission of these parasites compared with the other two ecological regions studied: the tropical, warm and humid climate throughout the year in the Amazon region is optimal for the development and survival of STH.[24 25]

In contrast, the infection rate of STH in the coastal region was relatively low compared with previous studies. Our results of a 14.5% prevalence for any STH differ from two studies conducted in the provinces of Manabi[26] and Esmeraldas,[27 28] where STH prevalence was between 65% and 74.9%, respectively, but with light or moderate infection intensities predominating. There are several possible explanations for these differences: (1) the present study was designed to select a representative sample of schoolchildren in coastal Ecuador while the previous studies were directed at populations considered to be at high risk of STH infections; (2) the present survey was done during 2011–2012 (after the introduction of systematic government-supported anthelmintic treatment programmes, initiated in 2006 and targeted at schoolchildren) while the previous studies were done before the introduction of such programmes and, therefore, might reflect the impact on prevalence of such school-based treatments; (3) differences in diagnostic methods used or the number of stools collected from each child will affect diagnostic sensitivity—the Manabí study,[26] that also used the Kato-Katz to detect STH, collected three serial stool samples for STH detection which will have increased diagnostic sensitivity compared with a single sample as collected in our study. Diagnostic sensitivity in the Esmeraldas study would have been increased by the use of a concentration method in addition to Kato-Katz[27]; and (4) the overall prevalence and percentage of high-intensity infections found in our study may have been underestimated because the majority of samples in this region were taken in the dry season when transmission is lower.

STH infections were of low prevalence (<20%) and light intensity in the highland region where environmental conditions for the development of these STH are not optimal. Most of the surveyed schools were located at altitudes between 1500 and 3000 m where there is low humidity and temperatures range between 10°C and 16°C, limiting the transmission of these parasites. A low prevalence of STH was found also in a high altitude rural community (*A. lumbricoides:* 20%; *T. trichiura*: 6.4%) with mostly mild to moderate infection intensities.[29] Other studies made in subtropical areas of the highlands have shown a higher STH prevalence,[30] but were not areas selected for study in the present survey.

The nutritional status of school-age children in the study was characterised by significant levels of malnutrition (14.2%) with stunting being most frequent (12.3%), particularly among schoolchildren from the Amazon region. These data are in agreement with the results of the National Survey in Health and Nutrition which found levels of growth retardation of 15% in children between 5 to 11 years in Ecuador,[23] In this national survey, most districts in the coastal and Amazon regions had a prevalence of stunting below 20%, whereas most districts of the highland region had stunting rates between 10% and 35% with the indigenous communities in the highlands being the most affected.[23] Our study observed a prevalence of stunting <20% in the highland region which is likely to be explained by the fact that our sample did not include predominantly indigenous populations.

Disease caused by STH is directly related to infection intensity.[31] We observed that children with moderate to heavy *A. lumbricoides* infections were 1.85 more likely to have malnutrition than children without infection. This positive association was mainly for growth retardation measured by HAZ used as an indicator of chronic malnutrition and is in accordance with previous studies indicating stunting to be a common consequence of STH.[32 33] There are several mechanisms by which STH could affect growth in children, including reduced food intake due to malabsorption and/or reduced appetite.[33] A study in Northeast Brazil showed that in a cohort of children aged 2–7 years, helminthiasis acquired in early childhood was associated with a 4.6 cm shortfall in height by the age of 7.[34] In addition, a meta-analysis indicated that where the prevalence of intestinal nematodes is ≥50%, anthelmintic drugs may have significant benefits including gains in weight, height, mid-upper arm circumference and skinfold thickness compared with untreated children.[4]

Because we selected representative samples in each of the three ecological zones, our findings are likely to be generalisable to schoolchildren of Ecuador and elsewhere in the Latin American region with similar levels of human development and geoclimatic characteristics. Based on our findings and in accordance with the WHO guidelines,[13] we classify schoolchildren from the Amazon region of Ecuador as being at high risk for STH infections (prevalence of any STH ≥50%) and recommend that all school-age children (enrolled and non-enrolled) in this region should be treated with anthelmintics twice a year. In coastal and highland regions (prevalence <20%), current indications recommend no large-scale preventive chemotherapy and treatment of affected individuals to be provided on a case-by-case basis. However, given that this survey was done in the context of previous interventions with anthelmintics, it would be prudent to continue annual treatments to prevent a potential rebound in prevalence until repeat surveys are able to confirm limited transmission in these regions. In addition, because previous studies have shown high prevalence areas in these two regions, we recommend that subtropical and tropical areas within these regions, where environmental conditions are propitious for STH transmission, should also be considered at high risk and twice yearly anthelmintic treatments given to schoolchildren until more detailed local surveys can be done to better define STH infection risk within each province. Long-term interventions that should be implemented in parallel with chemotherapy include improved sanitation and water supply and health education.

A limitation of this study is the nature of a cross-sectional study that does not allow us to determine the temporal relationship between the presence of STH infection and malnutrition. In addition, a single stool sample may underestimate STH prevalence, especially for hookworm infection. The data from this study will help health authorities in Ecuador to develop an operational plan for the treatment of preschool and schoolchildren in the three ecological regions in Ecuador. Clearly, efforts should be focused on the Amazon region where the prevalence is greater than 50% and where there are populations of marginalised and highly vulnerable groups such as indigenous groups. Similarly, these baseline data will allow health authorities to monitor the impact of control programmes. Although this study was carried out in 2011–2012, a national deworming programme has not yet been launched in Ecuador.

## CONCLUSIONS

We have estimated the prevalence of STH infections in the three ecological zones in Ecuador and observed the highest prevalence and intensity of infection in the Amazon region, likely a consequence of poor living conditions and an environment that is highly favourable for transmission. The prevalence of stunting was higher also in the Amazon region, and malnutrition was associated with the intensity of *A. lumbricoides* infections. There is a need for the implementation of deworming control programmes combined with interventions to improve nutrition for preschool and school-age children with a focus on the Amazon region. These programmes should be integrated with other existing programmes with the aim of reducing parasite burdens to prevent potential adverse effects on the nutritional and health status of children. Long-term strategies require improvements in drinking water, sanitation and hygiene.

**Acknowledgements** We thank the Lcda. Silvia Erazo (Centro de salud del IESS-El Batán), Lcda. Jackeline Fonseca (Hospital Provincial Docente Ambato) and Lcda. Gabriela Andrade (Hospital San Vicente de Paul-Ibarra) for their help with laboratory analyses; members of the Ecuadorian Ministry of Public Health and the Pan-American Health Organization for technical support; the Ecuadorian Ministry of Education for providing primary school databases and César Yumiseva for your help with the map. The school teachers, parents and children are thanked for their enthusiastic co-operation.

**Contributors** ALM, RL and PJC conceived and designed the study. ALM and RL collected data. ALM wrote the statistical analysis plan and cleaned and analysed the data. ALM and PJC wrote the paper. All authors approved the final version of the manuscript.

**Funding** Data collection was supported by Pan-American Health Organization. PJC was supported by Wellcome Trust grant 088862/Z/09/Z.

**Competing interests** None declared.

**Patient consent** Parental/guardian consent obtained.

**Ethics approval** The study protocol was approved by the Ethics Committee of the Universidad Central del Ecuador, Quito, Ecuador.

**Provenance and peer review** Not commissioned; externally peer reviewed.

**Data sharing statement** All data relating to the study are summarised in the article. Access to the original data can be obtained through the corresponding author.

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
