## [Reviewer comments · BMJ Open]

ARTICLE DETAILS

TITLE (PROVISIONAL)	Soil-transmitted helminth infections and nutritional status in Ecuador: findings from a national survey and implications for control strategies
AUTHORS	Moncayo, Ana-Lucia; Lovato, Raquel; Cooper, Philip

VERSION 1 – REVIEW

REVIEWER	Peter Steinmann Swiss Tropical and Public Health Institute, Basel, Switzerland
REVIEW RETURNED	24-Jan-2018

GENERAL COMMENTS	Moncayo and co-authors describe in their manuscript a national survey of soil-transmitted helminth (STH) prevalence and infection intensity among schoolchildren in Ecuador. They follow an established protocol and present the findings in a conventional way. The interpretation of the findings is adequate. The authors should be commended for a solid piece of work. Below, a couple of comments: Specific - Abstract: among the conclusions, deworming and health education are mentioned – what about the areas of sanitation and nutrition?- Strengths and limitations, and in the discussion section: considering the survey was completed a number of years ago, it would be interesting to know if any adaptations have been made to the national STH control program based on the findings of this survey.- Examination of stool samples: what type of larvae were intended to be detected on the Kato-Katz thick smear slides? Usually, no larvae are visible on such slides.- Statistical analysis: sewage – please be more specific- Characteristics: why was availability of piped water and sewers only assessed at school level and not at the level of the home of the study participants?- Prevalence and intensity: perhaps state the prevalence of moderate and high-intensity infections together – the aim of the intervention is to reduce to zero the presence of moderate and heavy-intensity infections. Another point: establishing an age-prevalence relationship is not very informative given the small range of age groups represented in the sample.- Association: please review the first sentence of the second paragraph.- Discussion: this is not a real baseline since interventions have already been implemented for several years. Related to that: the recommendation to switch to case-specific treatment needs to be reconsidered: switching from mass drug administration (MDA) to case-based treatment is only recommended once it has been established that transmission has been reduced sufficiently so as to ensure that the prevalence does not rebound after discontinuation of
---

	MDA.  - Conclusions: Amazon or Amazonian? - Table 2 and in text: the total/national estimates should be weighted by population fraction. As it is stated now, the average is heavily influenced by the results from the Amazon region which however only represents a small fraction of the national population. - Table 3: review title - Figure 2: it is impossible to distinguish the green denoting negative from the green associated with T. trichiura. - There are a couple of issues related to grammar – please carefully copy-edit the manuscript.
--	---

REVIEWER	Ana Sanchez, PhD. Professor. Department of Health Sciences Brock University Canada.
REVIEW RETURNED	24-Jan-2018

GENERAL COMMENTS	Date: January 24, 2018 Re: Manuscript review Journal: BMC Open Title: Soil-transmitted helminth infections and nutritional status in Ecuador: findings from a national survey and implications for control strategies By Moncayo AL, Lovato R and Cooper P. This a cross-sectional study investigating soil-transmitted helminth (STH) infection prevalence and its potential association with children's nutritional status in 3 ecological regions in Ecuador: Andes highlands, coastal lowlands, and Amazon plains. The sample size consisted of 920 schoolchildren and 6-16 years of age enrolled in 18 rural schools. Sample was stratified to obtain national representation. Results are as expected. They indicate a general STH prevalence of 27.9% (19.3% A. lumbricoides, 18.5% T. trichiura, and 5% hookworm 5%. However, children living in the Amazon were worse off in terms of both infections and malnutrition. Overall malnutrition 14.1% (stunting 12.3%). In general, a statistically significant association between A. lumbricoides and stunting was found. [OR 3.70 (1.48-9.24) p = 0.005] COMMENTS AND SUGGESTIONS ABSTRACT: a) The conclusion statement about the need for health education strategies is not incorrect but has nothing to do with the study. Conclusions should be based on results and their analysis. Perhaps something can be done to improve nutrition? STRENGTHS AND LIMITATIONS a) The first bulleted point “these data are from the first national survey” gives the impression that authors did a secondary data analysis. Revise accordingly. b) The third bulleted point “The sampling strategy may have not detected localized geographic areas within the three regions with a higher prevalence of STH infections” implies that if authors look
--

	further, they would have found more infections. But the opposite can also be true, can it? c) The fifth bulleted point should be revised “A single stool sample may underestimate STH prevalence” as it may be accurate for hookworm but no so much for Ascaris and Trichuris if the Kato-Katz is done well. d) Another limitation of the study is not doing a complete nutritional study but only anthropometry. Hematological blood workup and micronutrients determination. This should be added and discussed INTRODUCTION PAGE 5 = authors assert that the lack of data since 2009 indicates a lack of progress in the implementation of mass deworming campaigns. However, their data actually show that deworming treatment is given in the schools (Table 1). As well, data might not be reported to WHO/PAHO and therefore is missing from PCT-databank. If the assertion is true, it needs to be referenced to the Ministry of Health. METHODS Study area In addition to geographic and climatic description, a summary of the socioeconomic differences between the 3 zones is necessary. Every region is composed by provinces for which a human development index should exist. Literacy and proportion of children that go to school is also important. Equally important would be issues of food security for each region. As data shown, the condition in the Amazon are considerably different and this is one of the most important unaccounted confounders. It is as important to consider research participants that were left out of the study as much as those who volunteered. Study design and sample size a) Explain the ethical process as conducted in the field. The description gives the impression that children were told to participate. Data Collection b) Explain how demographic and other data were collected. Was it by observation by researchers? Face-to-face interview? If so, who provided responses and what is the reliability of the respondent (e.g., a child may be less reliable that an adult) Examination of stool samples a) Instead of the term “subjects” consider using “research participants” or “children” b) How long was the time from KK smear preparation until observation? This may alter the results, especially for hookworm c) Give here the categories for infection as opposed to do so in Table 2, which is already too busy Anthropometric measurements Why didn't the authors calculate WHAZ in the age group 7-10? Statistical analysis a) The statement “In multivariate analysis, the effect of geohelminth infections on malnutrition” although maybe “technically correct” leads to confusions since in a cross-sectional study we cannot determine the direction of causality. It would be better to
--	--

	rephrase and talk about the association of STH with malnutrition b) Is there an explanation why age was not used as a continuous variable? Using age groups when having such small range may cause more noise and information might be lost. c) It's very important to clarify if the cluster effect was by school. If so, why not by ecological region, since they are so very different? I believe that Stata doesn't allow to choose two variables to control for cluster, does it? And by the same token, why not do the regression analysis by ecological region? This would be more informative for intervention purposes. DATA PRESENTATION a) Table 2 is too busy and hard to read. Perhaps it can be formatted landscape and put the Cis on the same line. Infection categories with EPG could be better placed in the methodology. b) Figure 1. With such large error bars, the information is not very useful. c) Table 4 – Intensity of infection: you cannot have 4 categories for A. lumbricoides (none/light/moderate/heavy) and have 3 categories for the other two parasites (by merging moderate & heavy) just because of the small numbers. Choose either but not both. Would you lose significance if merged for A. lumbricoides as well? [Expressed as OR 3.70 (1.48-9.24)] DISCUSSION a) In my opinion, the discussion is unnecessarily long considering the simplicity of the study. Socioeconomic information known a priori about the ecological areas would fit better under the section "study area". b) More information about Ecuador's deworming programs or lack thereof is necessary. c) Recommendations in terms of STH are reasonable but the nutritional aspects of the study seemed somewhat ignored. d) It would be interesting to know how the situation in Ecuador compares with other countries in the region and/or Mexico and Central America.
--	--

VERSION 1 – AUTHOR RESPONSE

Point-by-point response to reviewers' comments

Reviewer: 1

Reviewer Name: Peter Steinmann

Institution and Country: Swiss Tropical and Public Health Institute, Basel, Switzerland

Please state any competing interests: None declared

Please leave your comments for the authors below

Moncayo and co-authors describe in their manuscript a national survey of soil-transmitted helminth (STH) prevalence and infection intensity among schoolchildren in Ecuador. They follow an established protocol and present the findings in a conventional way. The interpretation of the findings is adequate. The authors should be commended for a solid piece of work. Below, a couple of comments:

Specific

C: Abstract: among the conclusions, deworming and health education are mentioned – what about the areas of sanitation and nutrition?

R: Modified as suggested

C: Strengths and limitations, and in the discussion section: considering the survey was completed a number of years ago, it would be interesting to know if any adaptations have been made to the national STH control program based on the findings of this survey.

R: Text was added to Discussion section to read: “Although this study was carried out in 2011-2012, a national deworming programme has not yet been launched in Ecuador”.

C: Examination of stool samples: what type of larvae were intended to be detected on the Kato-Katz thick smear slides? Usually, no larvae are visible on such slides.

R: We agree with the reviewer. “Larvae” was eliminated from the text

C: Statistical analysis: sewage – please be more specific

R: Modified as suggested

C: Characteristics: why was availability of piped water and sewers only assessed at school level and not at the level of the home of the study participants?

R: The aim of the study was to obtain baseline data of the prevalence and intensity of STH infections to establish the deworming strategy in the country and not to determine the risk factors for infection at an individual level. Data on piped water and sewage were collected only for schools.

C: Prevalence and intensity: perhaps state the prevalence of moderate and high-intensity infections together – the aim of the intervention is to reduce to zero the presence of moderate and heavy-intensity infections. Another point: establishing an age-prevalence relationship is not very informative given the small range of age groups represented in the sample.

R: We show now the prevalence of moderate and heavy intensity infections combined in Table 2 and state these findings in the text of the Results. We consider it informative to provide disaggregated information regarding intensity of infection but have made the Figure 1 a supplementary file for referral to interested readers and have shortened the text of the results referring to these observations

C: Association: please review the first sentence of the second paragraph.

R: Sentence was reviewed and corrected

C: Discussion: this is not a real baseline since interventions have already been implemented for several years. Related to that: the recommendation to switch to case-specific treatment needs to be reconsidered: switching from mass drug administration (MDA) to case-based treatment is only recommended once it has been established that transmission has been reduced sufficiently so as to ensure that the prevalence does not rebound after discontinuation of MDA.

R: Our recommendations were based on The World Health Organization (WHO) Guideline: “Helminth Control in school-age children: a guide for managers of control programmes” and Preventive Chemotherapy to Control Soil-Transmitted Helminth Infections in At-Risk Populations Groups. WHO recommends preventive chemotherapy, using annual single-dose albendazole (400mg) or mebendazole (500mg) as a public health intervention for all school-age children living in areas where the baseline prevalence of any soil-transmitted infection is 20% and greater. Biannual administration is recommended where the baseline prevalence is over 50%. However, we agree with the argument of the reviewer in the context of regions where some form of intervention has preceded a survey, to define the intervention required. We have, therefore, changed the text to, “However, given that that this survey was done in the context of previous interventions with anthelmintics, it would be prudent to

continue annual treatments to prevent a potential rebound in prevalence until repeat surveys are able to confirm limited transmission in these regions.”

C: Conclusions: Amazon or Amazonian?

R: text modified to “Amazon region”

C: Table 2 and in text: the total/national estimates should be weighted by population fraction. As it is stated now, the average is heavily influenced by the results from the Amazon region which however only represents a small fraction of the national population.

R: The analysis follows the WHO guideline “Helminth Control in school-age children: a guide for managers of control programmes” to allow comparison with other international studies.

However, we have added national prevalence weighted by number of children aged 5-14 years old in rural areas of each region in Section “Prevalence and Intensity of STH infection”.

A paragraph was also added to section “Statistical analysis”

C: Table 3: review title

R: The title of table 3 has been modified for greater clarity.

C: Figure 2: it is impossible to distinguish the green denoting negative from the green associated with *T. trichiura*.

R: Modified as suggested. Figure 2 has now been relabeled Figure 1 and the previous Figure 1 moved to supplementary files as Figure S1.

C: There are a couple of issues related to grammar – please carefully copy-edit the manuscript.

R: Text was reviewed as suggested

Reviewer: 2

Reviewer Name: Ana Sanchez, PhD. Professor.

Institution and Country: Department of Health Sciences, Brock University, Canada.

Please state any competing interests: None to declare.

Please leave your comments for the authors below

Date: January 24, 2018

Re: Manuscript review

Journal: BMC Open

Title: Soil-transmitted helminth infections and nutritional status in Ecuador: findings from a national survey and implications for control strategies

By Moncayo AL, Lovato R and Cooper P.

This a cross-sectional study investigating soil-transmitted helminth (STH) infection prevalence and its potential association with children’s nutritional status in 3 ecological regions in Ecuador: Andes highlands, coastal lowlands, and Amazon plains.

The sample size consisted of 920 schoolchildren and 6-16 years of age enrolled in 18 rural schools. Sample was stratified to obtain national representation.

Results are as expected. They indicate a general STH prevalence of 27.9% (19.3% *A. lumbricoides*, 18.5% *T. trichiura*, and 5% hookworm 5%. However, children living in the Amazon were worse off in terms of both infections and malnutrition. Overall malnutrition 14.1% (stunting 12.3%). In general, a

statistically significant association between *A. lumbricoides* and stunting was found. [OR 3.70 (1.48-9.24) $p = 0.005$]

COMMENTS AND SUGGESTIONS

ABSTRACT:

C: a) The conclusion statement about the need for health education strategies is not incorrect but has nothing to do with the study. Conclusions should be based on results and their analysis. Perhaps something can be done to improve nutrition?

R: Conclusion statement was modified accordingly to read: "The implementation of school based preventive chemotherapy and nutritional supplement programmes within the Amazon region should be prioritized. Long-term control strategies require improvements in water, sanitation, and hygiene".

STRENGTHS AND LIMITATIONS

C: a) The first bulleted point "these data are from the first national survey" gives the impression that authors did a secondary data analysis. Revise accordingly.

R: The statement was modified to read: "This study is the first national survey of STH infections in schoolchildren"

C: b) The third bulleted point "The sampling strategy may have not detected localized geographic areas within the three regions with a higher prevalence of STH infections" implies that if authors look further, they would have found more infections. But the opposite can also be true, can it?

R: There is some variation in altitude and climate within each region of Ecuador, specifically in Andean region where is possible to find subtropical areas. Therefore, if we have selected schools by altitude or climate zones within each region, it is likely that have found children with higher prevalence of STH infections.

C: c) The fifth bulleted point should be revised "A single stool sample may underestimate STH prevalence" as it may be accurate for hookworm but no so much for *Ascaris* and *Trichuris* if the Kato-Katz is done well.

R: Text was modified to read: "A single stool sample may underestimate STH prevalence, especially or hookworm infection".

C: d) Another limitation of the study is not doing a complete nutritional study but only anthropometry. Hematological blood workup and micronutrients determination. This should be added and discussed

R: The primary aim of the study was to obtain baseline data of the prevalence and intensity of STH infections in order to establish the deworming strategy in the country. However, we also decided to collect anthropometric measurements. A complete nutritional evaluation was not performed for reasons of cost. Text was added to read: "Lack of a more complete nutritional evaluation including micronutrients"

INTRODUCTION

C: PAGE 5 = authors assert that the lack of data since 2009 indicates a lack of progress in the implementation of mass deworming campaigns. However, their data actually show that deworming treatment is given in the schools (Table 1). As well, data might not be reported to WHO/PAHO and therefore is missing from PCT-databank. If the assertion is true, it needs to be referenced to the Ministry of Health.

R: The reviewer is correct in that 84.3% of children had received anthelmintic treatment in the past year in schools. However, there is a lack of data at WHO/PAHO and the Ecuadorian Ministry of Public

Health (who were consulted directly with respect to this) on treatment coverage of pre-school and school children. We have edited the statement to read, "However, since 2009, there are no available data on treatment coverage of preschool and school-age children."

METHODS

C: Study area

In addition to geographic and climatic description, a summary of the socioeconomic differences between the 3 zones is necessary. Every region is composed by provinces for which a human development index should exist. Literacy and proportion of children that go to school is also important. Equally important would be issues of food security for each region. As data shown, the condition in the Amazon are considerably different and this is one of the most important unaccounted confounders. It is as important to consider research participants that were left out of the study as much as those who volunteered.

R: Text was added to read: "Poverty based on Unsatisfied Basic Needs was higher in the Amazon region (51.7%) than in the other regions (Coastal region: 40.1% and Andean region: 20.7%) in 2016. Net enrollment on basic education and illiteracy rates was higher in Andean region (97.4%, 5.9%) than in Coastal (95.4% ,5.5%) and Amazon regions (95.7%, 5.0%)."
Participation rates for the survey were added in "Characteristics of study population" to read: "A total of 938 children from the three regions and 18 schools studied were eligible of which 920 (98.0%) were recruited and provided stool samples.".

Study design and sample size

C: a) Explain the ethical process as conducted in the field. The description gives the impression that children were told to participate.

R: Ethics section was added to read: "The study protocol was approved by the ethics committee of the Universidad Central del Ecuador. Written informed consent was obtained from the parent of each child and verbal assent from each child to participate. All children were offered appropriate treatment".

Data Collection

C: b) Explain how demographic and other data were collected. Was it by observation by researchers? Face-to-face interview? If so, who provided responses and what is the reliability of the respondent (e.g., a child may be less reliable than an adult)

R: Text added to read: "Information on water supply, sanitation and anthelmintic treatment in schools was collected from teachers through an investigator-administered questionnaire and confirmed by direct observation"

Examination of stool samples

C: a) Instead of the term "subjects" consider using "research participants" or "children"

R: Modified as suggested

C: b) How long was the time from KK smear preparation until observation? This may alter the results, especially for hookworm

R: Text was added to read: "Kato-Katz slides were examined within 30-60 minutes of preparation"

C: c) Give here the categories for infection as opposed to do so in Table 2, which is already too busy

R: Modified as suggested.

Anthropometric measurements

C: Why didn't the authors calculate WHAZ in the age group 7-10?

R: "Weight-for-age is inadequate indicator for monitoring child growth beyond pre-school years due to its inability to distinguish between relative height and body mass, therefore, BMI-for-age is recommended by the WHO to assess thinness/wasting in school-aged children and adolescents".¹

¹ De Onis, M., Onyango, A. W., Borghi, E., Siyam, A., Nishida, C., & Siekmann, J. (2007). Development of a WHO growth reference for school-aged children and adolescents. *Bulletin of the World Health Organization*, 85(9), 660–667. <http://doi.org/10.2471/BLT.07.043497>.

Statistical analysis

C: a) The statement "In multivariate analysis, the effect of geohelminth infections on malnutrition" although maybe "technically correct" leads to confusions since in a cross-sectional study we cannot determine the direction of causality. It would be better to rephrase and talk about the association of STH with malnutrition

R: Modified as suggested

C: b) Is there an explanation why age was not used as a continuous variable? Using age groups when having such small range may cause more noise and information might be lost.

R: Age variable was used as a control variable in the model. The introduction of age as categorical or continuous variable in the model did not change the results.

C: c) It's very important to clarify if the cluster effect was by school. If so, why not by ecological region, since they are so very different? I believe that Stata doesn't allow to choose two variables to control for cluster, does it? And by the same token, why not do the regression analysis by ecological region? This would be more informative for intervention purposes.

R: The model we used considered only clustering by school because data correlation by region was low (*A. lumbricoides*: $\rho = 0.33$ and *T. trichiura*: $\rho = 0.31$). Adjustment of models by region did not materially affect the estimates of effect. For example, the OR for the association between moderate to heavy intensity of *Ascaris* infection and malnutrition adjusted by region was identical as when adjusted for clustering by school (OR 1.85, CI 95% 1.03-3.32).

DATA PRESENTATION

C: a) Table 2 is too busy and hard to read. Perhaps it can be formatted landscape and put the Cis on the same line. Infection categories with EPG could be better placed in the methodology.

R: Modified as suggested

C: b) Figure 1. With such large error bars, the information is not very useful.

R: Figure 1 has now been moved to supplementary files.

C: c) Table 4 – Intensity of infection: you cannot have 4 categories for *A. lumbricoides* (none/light/moderate/heavy) and have 3 categories for the other two parasites (by merging moderate & heavy) just because of the small numbers. Chose either but not both. Would you loose significance is merged for *A. lumbricoides* as well? [Expressed as OR 3.70 (1.48-9.24)]

R: We agree with the reviewer. We have now included moderate and heavy intensity of *Ascaris* infection as a single category in the Table.

DISCUSSION

C: a) In my opinion, the discussion is unnecessarily long considering the simplicity of the study. Socioeconomic information known a priori about the ecological areas would fit better under the section "study area".

R: Modified as suggested

C: More information about Ecuador's deworming programs or lack thereof is necessary.

R: This information was shown in introduction section and text was added in discussion section to read: "Although this study was carried out in 2011-2012, a national deworming programme has not been launched in Ecuador based on the findings of this survey".

C: c) Recommendations in terms of STH are reasonable but the nutritional aspects of the study seemed somewhat ignored.

R: Conclusion was modified accordingly to read: "There is a need for the implementation of deworming control programs combined with interventions to improve nutrition for pre-school and school-age children with a focus on the Amazon region. These programs should be integrated with other existing programs with the aim of reducing parasite burdens to prevent potential adverse effects on the nutritional and health status of children. Long-term strategies require improvements in drinking water, sanitation, and hygiene

C: d) It would be interesting to know how the situation in Ecuador compares with other countries in the region and/or Mexico and Central America.

R: Text was added to Discussion sections to read: "A systematic review of studies of prevalence of soil-transmitted helminth infection in South America from 2005-2012 estimated a prevalence of 28.1% for Ecuador, similar to the prevalence found in our study. Prevalence rates below 20% were reported for Argentina (18.9%) and Uruguay (18.8%) whereas the highest prevalence rates were reported for French Guyana (46.2%) and Surinam (40.1%). The remaining countries showed prevalence rates of between 25% and 39%".

Responses to Editorial comments (05-03-2018)

Comment 1: Your 'Strengths and limitations' have exceeded its number, it should only have a minimum of three (3) up to five (5) bullet format that relate specifically to the study reported. This should be placed after the abstract.

Response 1: Strengths and limitations have been limited to 5 bullet points, placed after the abstract. The limitation of collecting a single stool samples has been moved to the Discussion (page 28).

Comment 2: Please ensure to have the same DATA SHARING STATEMENT both in your main document and in Scholar One.

Response 2: Done

Comment 3: We have implemented an additional requirement to all articles to include 'Patient and Public Involvement' statement within the main text of your main document. Please refer below for more information regarding this new instruction

Response 3: Done (page 9)

VERSION 2 – REVIEW

REVIEWER	Peter Steinmann Swiss TPH, Switzerland
REVIEW RETURNED	13-Mar-2018

GENERAL COMMENTS	The authors have adequately addressed the concerns expressed by this reviewer upon reviewing a prior version of this manuscript.
--

REVIEWER	Ana Sanchez, PhD. Professor. Department of Health Sciences Brock University St. Catharines, Ontario. Canada
REVIEW RETURNED	14-Mar-2018

GENERAL COMMENTS	Thank you for the revisions.
------------------------------